# Predicting the causative pathogen among children with pneumonia using a causal Bayesian network

Yue Wu[1,2]*, Steven Mascaro[3,4], Mejbah Bhuiyan[2], Parveen Fathima[1], Ariel O. Mace[2,5,6], Mark P. Nicol[7], Peter C. Richmond[2,5,8], Lea-Ann Kirkham[2], Michael Dymock[2], David A. Foley[2,9], Charlie McLeod[2,10], Meredith L. Borland[8,11], Andrew Martin[5], Phoebe C. M. Williams[1,12,13], Julie A. Marsh[2], Thomas L. Snelling[1,2,12,14,15], Christopher C. Blyth[2,8,9,10]

1 Sydney School of Public Health, University of Sydney, Camperdown, New South Wales, Australia, 2 Wesfarmers Centre of Vaccines and Infectious Diseases, Telethon Kids Institute, University of Western Australia, Nedlands, Western Australia, Australia, 3 Bayesian Intelligence Pty Ltd, Upwey, Victoria, Australia, 4 Faculty of Information Technology, Monash University, Clayton, Victoria, Australia, 5 Department of General Paediatrics, Perth Children's Hospital, Nedlands, Western Australia, Australia, 6 Department of Paediatrics, Fiona Stanley Hospital, Murdoch, Western Australia, Australia, 7 School of Biomedical Sciences, University of Western Australia, Crawley, Western Australia, Australia, 8 School of Medicine, University of Western Australia, Crawley, Western Australia, Australia, 9 Microbiology, PathWest Laboratory Medicine QEII Medical Centre, Nedlands, Western Australia, Australia, 10 Infectious Diseases Department, Perth Children's Hospital, Nedlands, Western Australia, Australia, 11 Emergency Department, Perth Children's Hospital, Nedlands, Western Australia, Australia, 12 Sydney Children's Hospitals Network, New South Wales, Australia, 13 School of Women's and Children's Health, The University of New South Wales, Kensington, New South Wales, Australia, 14 School of Public Health, Curtin University, Bentley, Western Australia, Australia, 15 Menzies School of Health Research, Charles Darwin University, Darwin, Northern Territory, Australia

* yue.wu1@sydney.edu.au

## Abstract

### Background

Pneumonia remains a leading cause of hospitalization and death among young children worldwide, and the diagnostic challenge of differentiating bacterial from non-bacterial pneumonia is the main driver of antibiotic use for treating pneumonia in children. Causal Bayesian networks (BNs) serve as powerful tools for this problem as they provide clear maps of probabilistic relationships between variables and produce results in an explainable way by incorporating both domain expert knowledge and numerical data.

### Methods

We used domain expert knowledge and data in combination and iteratively, to construct, parameterise and validate a causal BN to predict causative pathogens for childhood pneumonia. Expert knowledge elicitation occurred through a series of group workshops, surveys and one-on-one meetings involving 6-8 experts from diverse domain areas. The model performance was evaluated based on both quantitative metrics and qualitative expert validation. Sensitivity analyses were conducted to investigate how the target output is influenced

**Data Availability Statement:** Source models, evaluation results and scripts for conducting the evaluations have been made publicly available via Open Science Frame, https://osf.io/m97vb/.

Individual-level patient data is restricted for sharing due to ethical requirements and is stored on secure server at the Telethon Kids Institute (TKI), in accordance with the TKI Information Retention and Disposal Policy and the TKI Research Data Confidentiality Policy. If the researcher intends to access the data for verification purpose only, please contact TKI Governance Officer, ResearchGovernance@telethonkids.org.au. For any additional analysis/ secondary use of the data, an application will need to be made to an ethics committee, e.g., Child and Adolescent Health Service, Human Research Ethics Committee (CAHS HREC), at CAHS.Ethics@health.wa.gov.au.

**Funding:** This project was supported by Perth Children Hospital Foundation (PCHF, Grant ID 9900), including YW, SM, MB, JAM, MN, TLS and CCB. In addition, YW is supported by the Western Australian Health Translation Network Early Career Fellowship which was supported in part by the Australian Government's Medical Research Future Fund (MRFF). CCB is supported by an Investigator grant from the National Health and Medical Research Council (NHMRC, GNT1173163). TLS is supported by a NHMRC Career Development Fellowship (GNT1111657). LK is supported by a Perron Foundation Fellowship. AOM is supported by a NHMRC Postgraduate Scholarship (GNT1191465). CM is supported by a Raine Clinician Research Scholarship. PCMW is supported by a NHMRC Investigator grant (GNT1197335). The funders had no role in study design, data collection and analysis, decision to publish, or preparation of the manuscript. URLs: Perth Children Hospital Foundation, https://protect-au.mimecast.com/s/cy5BCvl1rKiW5MXn5Cz0rO2?domain=pchf.org.au Western Australian Health Translation Network, https://protect-au.mimecast.com/s/SJ7ICwV1vMfLmYpEmTKhgpU?domain=wahtn.org National Health and Medical Research Council, https://www.nhmrc.gov.au/ Perron Foundation, https://protect-au.mimecast.com/s/n9Z-CxngwOfJVrO5VfRbZ5b?domain=perronfoundation.org.au Raine Foundation, https://protect-au.mimecast.com/s/Fhx5CyojxQTNw9nPwSA8itN?domain=rainefoundation.org.au.

**Competing interests:** The authors have declared that no competing interests exist.

by varying key assumptions of a particularly high degree of uncertainty around data or domain expert knowledge.

## Results

Designed to apply to a cohort of children with X-ray confirmed pneumonia who presented to a tertiary paediatric hospital in Australia, the resulting BN offers explainable and quantitative predictions on a range of variables of interest, including the diagnosis of bacterial pneumonia, detection of respiratory pathogens in the nasopharynx, and the clinical phenotype of a pneumonia episode. Satisfactory numeric performance has been achieved including an area under the receiver operating characteristic curve of 0.8 in predicting clinically-confirmed bacterial pneumonia with sensitivity 88% and specificity 66% given certain input scenarios (i.e., information that is available and entered into the model) and trade-off preferences (i.e., relative weightings of the consequences of false positive versus false negative predictions). We specifically highlight that a desirable model output threshold for practical use is very dependent upon different input scenarios and trade-off preferences. Three commonly encountered scenarios were presented to demonstrate the potential usefulness of the BN outputs in various clinical pictures.

## Conclusions

To our knowledge, this is the first causal model developed to help determine the causative pathogen for paediatric pneumonia. We have shown how the method works and how it would help decision making on the use of antibiotics, providing insight into how computational model predictions may be translated to actionable decisions in practice. We discussed key next steps including external validation, adaptation and implementation. Our model framework and the methodological approach can be adapted beyond our context to broad respiratory infections and geographical and healthcare settings.

## Author summary

Pneumonia is a leading cause of hospital visits among young children. Doctors need to weigh a range of observations to make timely diagnostic and management decisions, including to differentiate bacterial from viral infections. This is a difficult task to achieve without support due to the complex interactions among relevant factors, and is a major driver of unnecessary antibiotic use. We used domain expert knowledge and data to create a causal Bayesian network (BN) model which depicts an integrated picture of how biological, epidemiological and clinical processes interact to form the challenge of diagnosing and managing children who present to hospital for pneumonia. The model can produce reliable and explainable quantitative inference to help distinguish viral from bacterial infections. We used a few examples to demonstrate how the BN can be used in clinical context, and discussed how computational model predictions may be translated to actionable decisions in practice. We discussed key next steps including the evaluation of the model using external data sets, and adaptation and implementation of the model in other settings. We also discussed how our model framework and the methodological approach can be adapted beyond our context to broad respiratory infections and geographical and healthcare settings.

## 1 Introduction

Pneumonia, infection of the lower airways, remains a leading cause of hospitalisation and death among young children worldwide [1, 2]. In recent decades, marked reductions in the burden of disease have occurred thanks to public health strategies including immunisation targeting *Haemophilus influenzae* type b, *Streptococcus pneumoniae*, and influenza virus [3, 4], however substantial morbidity and mortality remains. These interventions have led to a decline in the contribution of bacterial pathogens in particular, challenging the appropriateness of routine empiric antibiotic use for treating community-acquired pneumonia (CAP) [5]. Viruses are increasingly considered a major cause of paediatric pneumonia especially in high income settings, and children with viral infection are unlikely to benefit from antibiotic therapy [6, 7]; rather, unnecessary antibiotic use may lead to otherwise avoidable side effects, and be a driver of antimicrobial resistance [8, 9]. Despite this, antibiotics are still widely prescribed for children with pneumonia [5] possibly due to the difficulty in excluding bacterial infection as well as clinician concern regarding the potential consequences of under-treatment of bacterial pneumonia whether caused by 'typical' bacterial pathogens such as *Streptococcus pneumoniae* (pneumococcus), atypical bacteria such as *Mycoplasma pneumoniae* (mycoplasma), or bacterial super-infection following viral infection [10].

The aetiology of CAP in children has been studied extensively [11]. A number of clinical and laboratory factors have been shown to be associated with pneumonia in sick children [12–14], and many of these features (including cough, increased respiratory effort and raised inflammatory markers) are common in both bacterial and non-bacterial pneumonia. However, no single predictor exists that is sufficiently sensitive or specific to reliably differentiate bacterial (including bacterial-viral co-infection) from non-bacterial pneumonia [9]. Also, the performance of possible predictors may vary by context, e.g., in early versus late stage illness, or where the underlying prevalence of a pathogen varies [15]. The contribution of bacteria and viruses to pneumonia has been reported to vary by age, season, vaccine coverage, socioeconomic status, and across countries [6, 16, 17], although a recent review found evidence that viruses associated for a similar proportion of pneumonias across settings [7]. Isolation of pathogens directly from the lower respiratory tract is highly specific and may be sensitive, but sampling is typically invasive, challenging in children, and rarely indicated clinically and consequently rarely done [18]. Although less specific, nasopharyngeal samples are frequently used for extrapolating pneumonia aetiology because of ease of collection, however any pathogens detected in the nasopharynx might not be the actual cause of the pneumonia [19]. Due to the dynamic nature of disease epidemiology, non-specific clinical presentation and diagnostic limitations of laboratory tests, the timely and accurate identification of pathogens causing pneumonia remains an ongoing challenge.

Mathematical prediction models have been developed to aid clinical diagnosis, and existing approaches are usually based on quantifying associations between the target (e.g. bacterial pneumonia) and certain input variables in specified patient groups. Typical statistical regression methods are agnostic to underlying causal mechanisms and ignore complex interactions, such as those that occur between epidemiological, clinical, microbiological and immunological factors in pneumonia. Nevertheless, reasonable diagnostic performance has been achieved [20, 21]. However, the implementation and uptake of these models have been limited for several reasons. First, validation and clinical implementation of models are often challenged by missing data [22], such as biomarkers which might be useful predictors but may not be available in all cases to aid timely clinical decision-making [21]. Second, the models may not be transportable from the training context to other clinical contexts, because the predictive values of input variables are often driven by baseline prevalence [15], infection severity and testing techniques

employed in the validated study population. Third, association-driven predictions that lack causal explanation may not be accepted by their intended end-users (i.e., clinicians) [23]. The clinical implementation of any model for decision support is largely dependent on trust which, in turn, depends on end-users understanding how the model works.

Bayesian networks (BNs) may offer a solution to the challenges of predictive model design and implementation into clinical practice [24]. BNs have been used to clarify complex medical problem domains including diagnosing CAP and ventilator-associated pneumonia [25–27], by facilitating probabilistic and causal reasoning using directed acyclic graphs (DAGs) [28, 29]. The graphical representation of BNs facilitates the translation of knowledge from domain experts and that acquired from clinical datasets into a causal inference framework, with quantitative relationships between variables. Causal BNs (that is, BNs built to have a causal interpretation) provide probabilistic predictions that are transparent and potentially explainable [30, 31], i.e., representing relevant mechanistic pathways that lead to and can be intervened upon to change downstream effects, as well as to identify the most probable causes that led to some event. This is something that cannot be achieved in any non-causal (i.e., predictive) modelling approach [32]. The development of causal BNs relies on collaboration between domain experts and modellers to synthesize an understanding of the problem domain with the modelling technique, creating model outputs that are meaningful for clinical practice. BN model outputs can be applied to a variety of patient subgroups (such as specific age groups or geographic locations), and can robustly account for missing input variables like biomarker measurements from laboratory data. Our previous work has demonstrated how epidemiological, procedural and laboratory observations can be organised under a BN designed for predicting the causative pathogen in other childhood infections, and how such a model may be presented to aid clinical decision making [33, 34].

Here we present a causal BN model that depicts the pathophysiology of pneumonia in children, which could be used to help distinguish bacterial (including typical and atypical bacteria, and mixed viral-bacterial infections) from non-bacterial infections by predicting the causative pathogen for a given episode. The model was constructed, parameterised, and validated using both domain expert knowledge and data. We used data obtained from 230 children admitted to a tertiary hospital in Western Australia from 2015–2018 with pneumonia who underwent clinical, microbiological, immunological and radiological assessment [5, 35]. A diverse group of domain experts participated in the knowledge elicitation process. Based on numeric evaluation metrics, the resulting BN showed stable and good performance in predicting clinically-confirmed bacterial pneumonia as well as the detection of pathogens in the nasopharynx. The model also produced clinically meaningful outputs as assessed by domain experts via evaluation workshops and surveys. To demonstrate the potential utility of the model, we apply it to three representative clinical scenarios and we then discuss what work still needs to be done to achieve the goal of a decision support tool that improves patient management and encourages judicious use of antibiotics for paediatric pneumonia.

## 2 Methods

### 2.1 Ethics statement

Ethics approval was granted by the Child and Adolescent Health Service Human Research Ethics Committee (RGS2477) at Perth Children's Hospital. Written informed consent was obtained from the parent or legal guardian of each participant.

A BN model consists of two components, 1) a DAG that qualitatively describes how variables (nodes) interact with each other, and 2) a joint conditional probability distribution that quantitatively specifies how changes in each parent variable probabilistically drive changes in

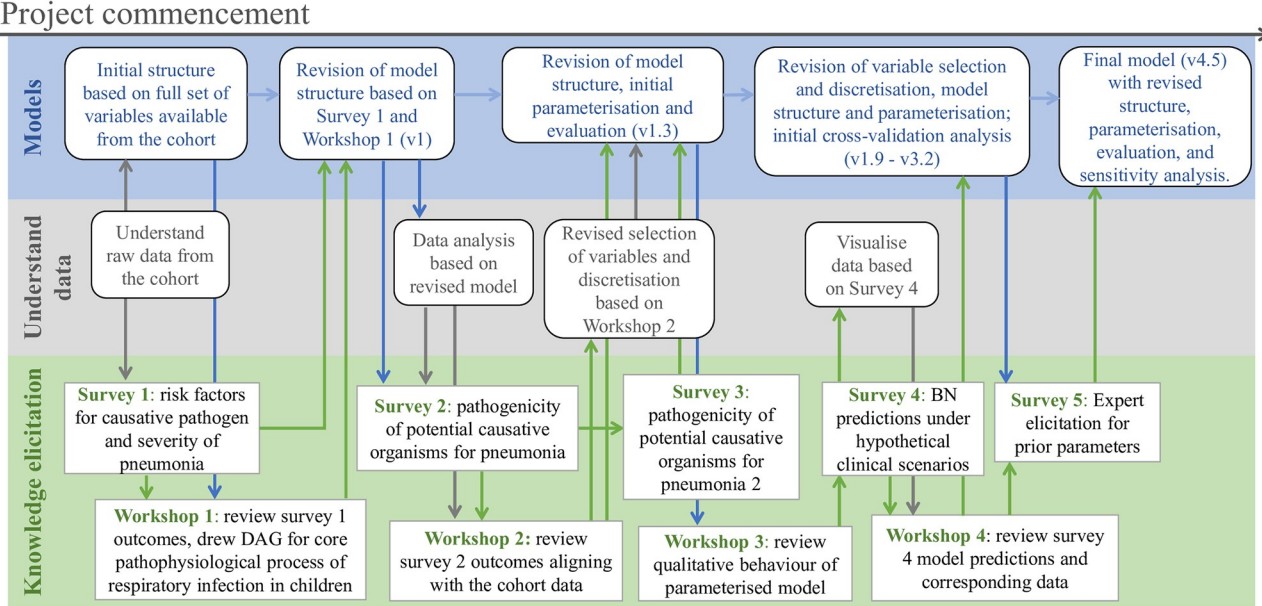

**Fig 1. The schematic of BN development.** The top blue row presents the development of the BN from its initial form (left) to the fourth iteration (right), and how this iteratively interacts with the data analysis (middle grey row) and the knowledge elicitation (KE, bottom green row). Everything is primarily designed to support model development; however the models themselves also support and drive data analysis and KE. In some cases, there are also direct interactions between data analysis and KE. During the KE process, survey outcomes often directly guide the design and focus of the following workshop. As the BN is developed and revised over several iterations, any data analysis, KE or modelling outcomes from an earlier iteration may inform any of these activities in the next iteration.

their child variables. The arrows (arcs) of the DAG indicate the presence of a direct influence of predecessor (or parent) variables on their child nodes (nodes extending from other nodes). A causal BN is one in which the arcs represent influences that are causal; sometimes with minor exceptions for associations that are required, but are not a core part of the causal process being modelled. Developing a causal BN involves constructing the model structure (the DAG) that describes the problem domain of interest, and quantifying the probabilistic effect that parent variables have on their child variables as a series of conditional probability tables (CPTs), i.e., parameterisation. In this project we used domain expert knowledge and data in combination and iteratively, to build a causal BN to predict causative pathogens for childhood pneumonia, aiming to distinguish bacterial from non-bacterial cases for which the former is more likely to benefit from antibiotic use. Fig 1 shows the schematic of the BN development process.

## 2.2 Knowledge elicitation

Expert knowledge elicitation occurred through a series of group workshops, surveys and one-on-one meetings involving 6–8 domain experts (with backgrounds in infectious disease, clinical microbiology, emergency medicine, general paediatrics, immunology and epidemiology). The theme of the workshop and survey was often driven by the evolving modelling progress—questions arising from data analysis, development of model structure and parameterisation. In total, as shown in Fig 1, four workshops were conducted covering the development of the model structure, data interpretation, model structure validation, and validation of the application use cases. Five surveys were distributed to inform variable selection, model validation via clinical use cases, and elicitation of key model parameters. One-on-one meetings were

organised when more detailed discussion was considered necessary by the modellers, such as to consolidate understanding of a particular topic discussed during a workshop. In S1 File, we provide the detailed description of the model development process (Section A, S1 File) associated with a full list of survey questions (Section B—F, S1 File).

## 2.3 BN structure and parameterisation

The final model structure was achieved through an iterative process of synthesising expert knowledge, data and learnings from earlier BN revisions. Parameterisaton of the model was conducted using the expectation maximisation (EM) algorithm [36], with one of the following three methods applied to each node:

A *data-driven* approach was used to derive CPTs directly from the observed distribution of relevant variables in the data, when data for those variables was sufficient.

An *expert-driven* approach was used for conceptual variables that were not or could not be recorded in the data. In this approach, the CPT was fully specified by domain expert knowledge alone and not further updated with data. For example, **susceptibility to progression** describes the extent to which an infected child is at risk of progressing from mild to severe manifestations of pneumonia. In the model, this susceptibility is higher in younger children, in those with chronic respiratory disease, and in those who are immunocompromised.

A *hybrid expert-data* approach was used in which expert-driven CPTs were used as a Bayesian prior (i.e., a starting point) assigned a weight of one data point and subsequently updated using available data. This method was preferred for variables that were strongly associated with other data, but could not be directly observed or mapped to any individual variable (i.e., latent), as well as for observable variables where data were sparse for certain patient subgroups. Note, the priors used can be accessed via the source model files on Open Science Framework (https://osf.io/m97vb/).

The parameterisation method for each variable was chosen based the definition of each variable and how well we were able to quantify them based on available data and/or expert knowledge. A description of the parameterisation method used for each variable is provided in the BN dictionary (S1 Table).

## 2.4 BN performance and evaluation

We evaluated the performance of the BN in two ways. Quantitatively, numerical evaluation ensured the model predictions were consistent with the observed data. While metrics that predict accuracy such as area under the receiver operating characteristic curve (AUROC) have been widely used, these measures are more suitable for definite classifications. Log loss (and similar probability-sensitive metrics) is considered more informative for evaluating models that generate probabilistic predictions [37] like BNs, because it rewards accurate probability estimates most highly. Log loss is calculated, for a given case, as the negative log of the model's estimated probability of what was actually observed (i.e., $-log\{P(observed\ outcome)\}$) with a lower log loss indicating a better fit of the model to the observed data. To quantitatively evaluate the model predictions from a clinical decision perspective, we also presented model predictions in terms of false positives (FP), false negatives (FN), their associated rates (FPR/FNR), sensitivity and specificity under different FPR/FNR trade-off preferences.

Sensitivity analyses were conducted using variance-based sensitivity analysis (VBSA) [38, 39], to investigate how the target variable of the model is influenced by varying the

assumptions for a range of key variables, especially those affected by a high degree of uncertainty around data or domain expert knowledge. More specifically, we simultaneously vary the parameters for the CPTs of a group of variables (by up to 20%) and observe how the distribution of a target variable state spreads given the variation in those parameters. Essentially, the VBSA allows us to investigate the impact on the target of the second order uncertainty that may exist in the CPT parameters (without incorporating that second order uncertainty directly into the model).

While acknowledging that overfitting cannot be properly determined until we can assess the model using external datasets, we can still gain some understanding of where overfitting may occur by identifying which part of the model is fitted most strongly to our cohort. We have adopted an approach that compares CPTs between the trained model and prior model using Kullback–Leibler (KL) divergence [40].

The approaches above are unfortunately of limited value in evaluating the causal features of a model. For example, it is common for two BNs to differ in their causal features but be identical in their statistical properties [29, 41]; such models would be assessed as equivalently good by any of the above metrics. More commonly, major differences in causal features may only lead to minor differences in predictive performance. Thus, while predictive performance can help identify significant issues (such as major missing pathways or dependencies), other approaches must be used to evaluate the individual causal features of a BN. This can be achieved in a number of ways (e.g., comparing to literature, conducting experiments, etc.); here we performed this validation with experts, by walking through the DAG structure afresh at each workshop, and by soliciting feedback on the DAG structure via followup emails. In addition, we performed qualitative expert validation of the model outputs, allowing us to assess if the model behaves in a way that meets clinician expectations and offers clinically informative predictions. We simulated three typical scenarios encountered in clinical practice, and presented BN predictions in both survey format (Survey 4) and during workshops (Workshops 3 and 4) for expert review.

We used the GeNIe BN (https://www.bayesfusion.com/) software to elicit the BNs presented here, and Netica (https://www.norsys.com/) to develop and parameterise subsequent models. Data, evaluation and sensitivity analyses were conducted using R [42] and Python [43]. Throughout the paper, we label the name of a BN variable in **bold**, and its state name in *italics* where referenced for the first time. Source models, raw evaluation results and python scripts for running the analysis can be accessed via Open Science Framework (https://osf.io/m97vb/).

## 3 Results

Fig 2 presents the high-level structure of the BN (v4.5), a model of paediatric pneumonia at the point of hospital presentation. We explicitly model the *existence* of infection separately from its clinical diagnosis (white nodes), for which the former is our primary target for prediction but often latent and cannot be directly observed, while the latter may be operationally defined by a set of clinical or laboratory observations and therefore can be used as a surrogate for evaluation. A more detailed description of the model is provided in §3.2.

### 3.1 Summary of model variables

The model consists of 62 variables including 14 which are background factors, six pathogen-specific variables describing the presence of pathogens in the nasopharynx, eight variables representing infection or the clinical diagnosis of infection, 25 variables representing specific signs and symptoms, five laboratory variables, and four intervention variables. In the BN

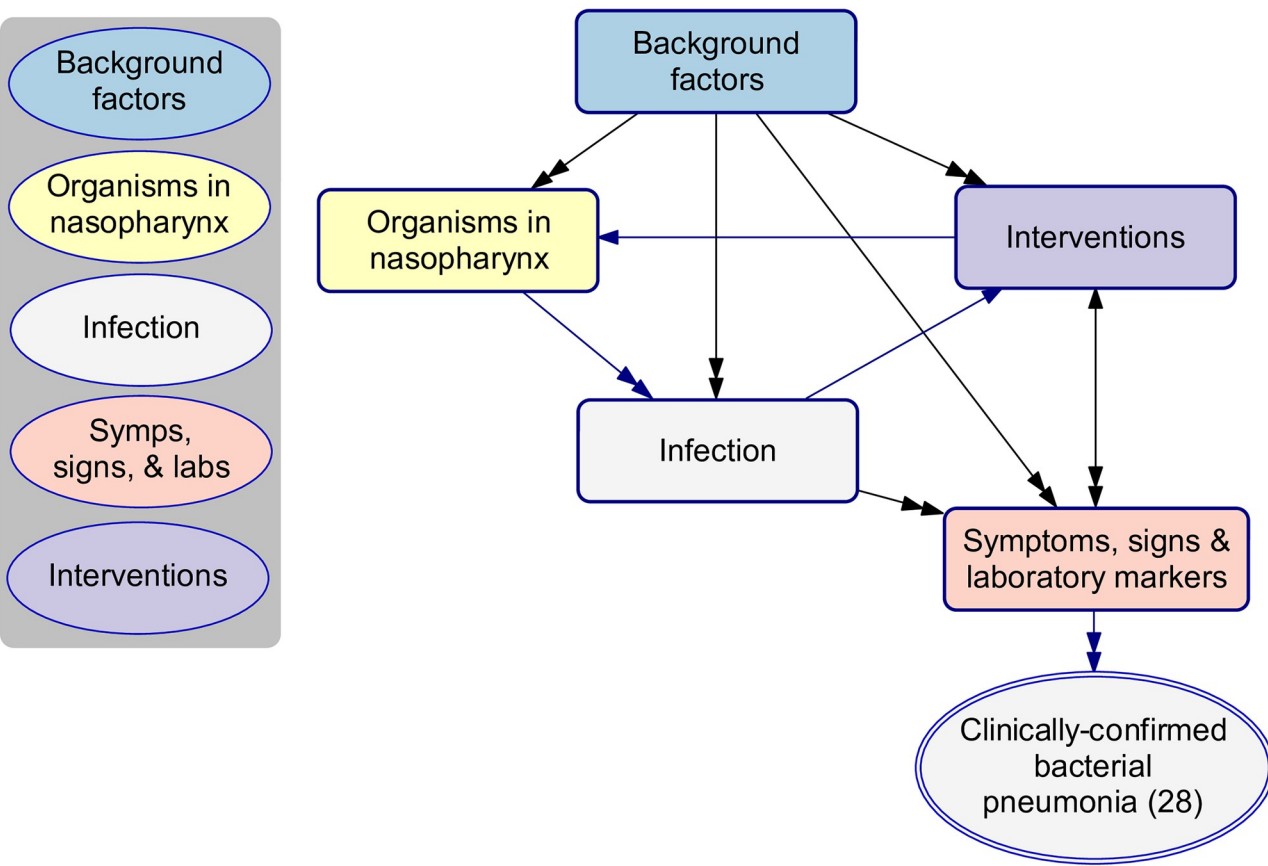

**Fig 2. The high-level model structure of paediatric pneumonia, BN v4.5.** Variables were organised into five highly interdependent groups (shown as rectangle boxes). In the centre of the diagram sits the core part of the model, infection of the respiratory tract (white box), which is predominantly influenced by the presence of a range of pneumonia-causing pathogens in a child's nasopharynx (yellow), as well as a number of background factors (blue). Infection manifests as clinical symptoms, signs and abnormal laboratory markers (salmon). Interventions (purple) refer to investigations and treatments which are relevant to the episode of pneumonia, and which may either occur before presentation (e.g., prior antibiotic use) potentially affecting variables like the culture result, or after presentation (e.g., supplemental oxygen) which may be affected by variables like the infection phenotype. S1 Table includes a BN dictionary which defines each variable.

dictionary, we provide information on each model variable, including identifiers (ID and variable name), what it means and how it's mapped to data (description), how it's discretised in the BN (states), its parents in the BN, and how it's affected (mechanistically rather than statistically) by those parent nodes. The dictionary also notes the observational status (whether observable, latent or derived), and the parameterisation method used (data-driven, expert-driven or hybrid expert-data) for each variable. Fig 3 shows a sample of the BN dictionary; the full dictionary is provided in S1 Table.

Alongside the 52 observable variables, nine latent variables were created for modelling purposes, including three concept variables that describe the epidemiological context (Fig 4, light blue nodes **(12)—(14)**) and six infection-relevant variables that describe the causative pathogens and the sites of the respiratory tract involved (Fig 5, white nodes **(21)—(26)**). These latent variables help to explicitly describe important underlying mechanisms that cannot be directly observed and thus captured by data. With respect to the choice of parameterisation method, 16 variables were exclusively trained from data (data-driven), 7 were exclusively parameterised based on expert knowledge (expert-driven), and 39 nodes were trained via the hybrid expert-

| ID | Variable name | Description | States | Parents in the BN | Relationship with parent nodes | Status | Parame-terisation | Category |
|---|---|---|---|---|---|---|---|---|
| 1 | Age group | Age group of study participant. In the model, we define each group as follow: Infant (<=2yo), … | *Infant, PreSchool, School* | None | Not applicable | Observable | Data-driven | Background factors |
| 14 | Susceptibility to progression | This describes the extent of the child to progress to more severe manifestation of pneumonia if infected. | *High, Low* | Age group, chronic respiratory disease, impaired immunity | The pneumonia is more likely to progress if the child's immune system is unable to clear the infection… | Latent | Expert-driven | Background factors |
| 21 | Viral-like nasopharyngeal infection | Replication of viral-like pathogens is occuring in the nasopharyngeal tissues. | *Present, Absent* | Organisms in nasopharyn (influenza, RSV, HMPV, parainfluenza, mycoplasma) | Presence of virus or mycoplasma in the nasopahrynx predisposes viral-like nasopharyngeal infection. | Latent | Expert-data | Infection |
| 44 | Crackles | Crackles/ crepitations as auscultatory finding recorded in the medical notes. | *Recorded, Unknown* | Current clinical phenotype, causative pathogen for pneumonia | Crackles refers abnormal lung sounds characterized by discontinuous clicking or … | Observable | Expert-data | Signs & symps |
| 55 | C-reactive proteins | The amount of C-reactive proteins (CRP) detected from blood. | *Above70, Btw30And70, Below30* | Current clinical phenotype, causative pathogen for pneumonia | Elevated level of CRP in blood can be driven by the systemic inflammatory response, which … | Observable | Expert-data | Labs |

**Fig 3. Extract from the BN dictionary.** The full dictionary is provided in S1 Table.

data approach. One variable (**current clinical phenotype (52)**) was introduced as a summary of the presenting clinical phenotypes of pneumonia. This variable was derived based on a separate clustering using the EM algorithm to determine a child's most probable pneumonia phenotype based on their clinical observations. The analysis resulted in two phenotypes, where *phenotype 1* represented a more severe type which increased the probability of all observable signs and symptoms compared to *phenotype 2*, and this information was added as an additional column in the cohort data and subsequently used as an observed variable. We discuss the role of this variable further below §3.2.3.

## 3.2 Model description

**3.2.1 The epidemiological context.** The model introduced three latent concepts (light blue) to simplify and describe the complex nature of pneumonia epidemiology in children; namely the **level of exposure (12)**, the **susceptibility to colonisation (13)**, and the **susceptibility to progression (14)** (Fig 4). The susceptibility to colonisation summarises the level of a host's susceptibility to being colonised in the nasopharynx with 'typical' bacterial pathogens, which is higher in the younger **age group (1)**, in children with **impaired immunity (6)**, and when there is a **smoker in the household (3)**. The level of exposure refers to the host's exposure to transmissible pathogens. Greater exposure is associated with older age, more **childcare days per week (5)**, and *Indigenous* Australian **ethnicity (2)**, which may in turn be surrogates of frequent and/or close or prolonged social contact and therefore greater opportunity for transmission. The susceptibility to progression describes the propensity of the host to pneumonia progression if infected, which is higher if the child is younger, is reported to have impaired immunity or has **chronic respiratory disease (7)**.

This BN explicitly describes the presence of six pathogens in the nasopharynx: influenza, respiratory syncytial virus (RSV), human metapneumovirus (HMPV), parainfluenza, mycoplasma and typical bacteria (e.g., pneumococcus) (Fig 4, yellow nodes **(15)—(20)**). Various epidemiological risk factors interact resulting in varied prevalence of each pathogen in the nasopharynx, which predispose children to respiratory tract infection including pneumonia, leading to the core part of the model presented in Fig 5. For example, while seasonality can influence the prevalence of many pathogens, in this model we use **influenza season (9)** (defined as June to September based on the expected pattern of influenza in the southern

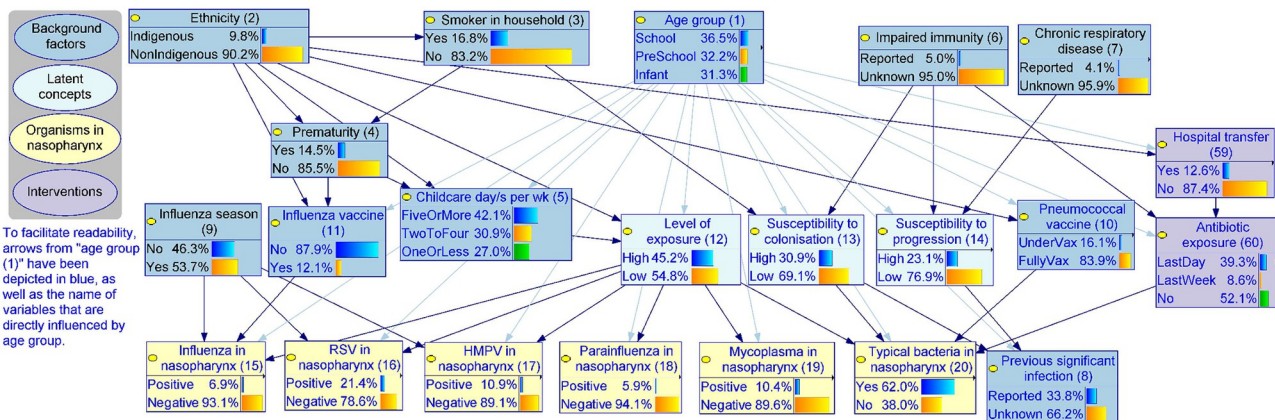

**Fig 4. The epidemiological context, BN v4.5.** Due to the number of arcs emanating from Age group, a lighter shade of blue has been used to make the graph easier to read. The detailed model structure and definition for each variable are provided in S1 Table. Source model files can be accessed via Open Science Framework (https://osf.io/m97vb/).

hemisphere in the pre-COVID-19 era) as a surrogate of the temporal increase in the prevalence of influenza, RSV and HMPV, with the model estimating increases of 220%, 173% and 177%, respectively. **Influenza vaccine (11)** reduces the probability that influenza is present in the nasopharynx by a modelled estimate of 67%. Recent **antibiotic exposure (60)** reduces the probability of typical bacteria being present in the nasopharynx from 75% to 49%.

**3.2.2 The infection.** Fig 5 shows how the BN depicts infection, based on whether viruses or typical or atypical bacteria are involved in a particular pneumonia episode, and whether upper respiratory sites (nasopharynx and/or throat) are involved when pneumonia is present. Mycoplasma is an intracellular bacterial pathogen with similarities to viruses with respect to mechanism of acquisition and some clinical manifestations. For modelling purposes, we grouped it with the viruses (described as the *viral-like* group) while keeping all the other bacteria in the *typical bacterial* group. For viral-like pathogens, the BN assumes infection of the lung (**viral-like pneumonia (23)**) can either occur directly without observed upper respiratory tract infection (URTI), or it can occur following observed URTI including **viral-like nasopharyngeal infection (21)** and **viral-like throat infection (22)**. The exact probability of each virus causing a respiratory infection is influenced by its pathogenicity (parameters estimated via the hybrid expert-data approach). Viral infection of the nasopharynx or throat increases the probability of viral pneumonia. Unlike viral-like pathogens, typical bacteria frequently colonise in the nasopharynx (62% in the model), described by **Typical bacteria in nasopharynx (20)**, and spontaneously cause pneumonia (**typical bacterial pneumonia (24)**) at a much lower frequency than viruses. In addition, pneumococcal vaccination, the presence of viral-like infection, and the age of the child, can all directly influence the probability of typical bacterial pneumonia.

We introduced the **causative pathogen for pneumonia (25)** and **upper airway involvement (26)** variables to summarise the above. We defined the causative pathogen for pneumonia to be *typical bacterial* if **typical bacterial pneumonia (24)** is *present*, *viral-like* if **viral-like pneumonia (23)** is *present* and typical bacterial pneumonia is *absent*, or *no pneumonia* if both the typical bacterial and viral-like pneumonia are *absent*. These *no pneumonia* cases can be other lower respiratory tract infections, like bronchitis, and about 7% cases were classified into this category. For our application's purpose of guiding antibiotic use, we force the model to re-

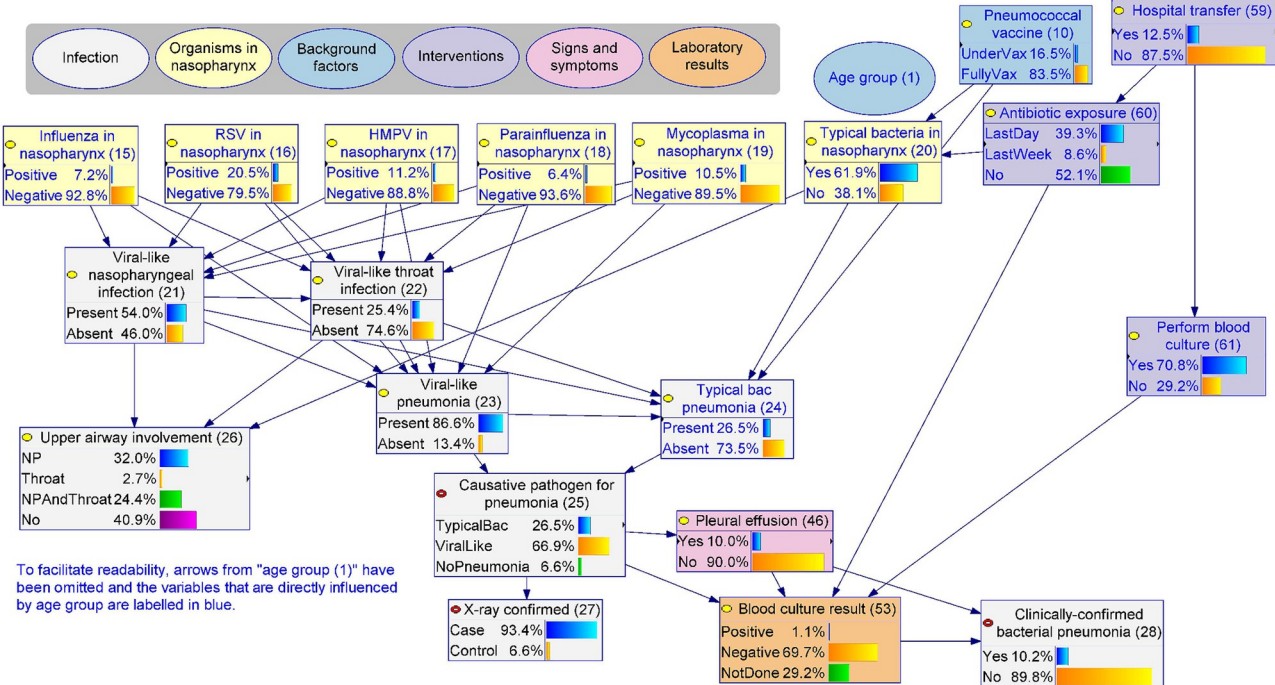

**Fig 5. The infection, BN v4.5.** The detailed model structure and definition for each variable are provided in S1 Table. Source model files can be accessed via Open Science Framework (https://osf.io/m97vb/).

classify these 7%, distributing them across typical bacterial or viral-like categories by setting **X-ray confirmed pneumonia (27)** to *case*. The operational definition of **clinically-confirmed bacterial pneumonia (28)** in the cohort study was either having a **pleural effusion (46)** or a *positive* **blood culture result (53)**, comprising 11% of children in the cohort. As illustrated in Fig 5, this definition is subject to potential measurement and selection biases introduced by prior antibiotic exposure, the decision to **perform blood culture (61)** and **hospital transfer (59)**.

The BN predicted that of all pneumonia presentations, 28.4% involved infection with typical bacterial pathogens with a mean of 25.2% based on the 10-fold cross validation (10-fold mean), while 92.8% (10-fold mean 94.6%) involve infection with viruses or viral-like pathogens. Among all pneumonia presentations involving infection with viruses or viral-like pathogens, 22.8% (10-fold mean 20.1%) were predicted to also involve infection with typical bacterial pathogens. From the decision-support perspective, it is important to differentiate infection with mycoplasma from viruses, and this can be addressed by combining information provided by two nodes, the **causative pathogen for pneumonia (25)** and **mycoplasma in nasopharynx (19)**. In other words, if the model predicts a high probability of infection with a *viral-like* pathogen and also a high probability of mycoplasma in nasopharynx, there is a high probability that mycoplasma is the causative pathogen (with possible implications for management).

**3.2.3 The evidence.** From a clinical perspective, a child's **causative pathogen for pneumonia (25)**, **current clinical phenotype (52)** and **upper airway involvement (26)** are not directly observed (i.e., they are latent), and can only be inferred from clinical observations and the results of laboratory investigations. From a causal perspective, these latent variables

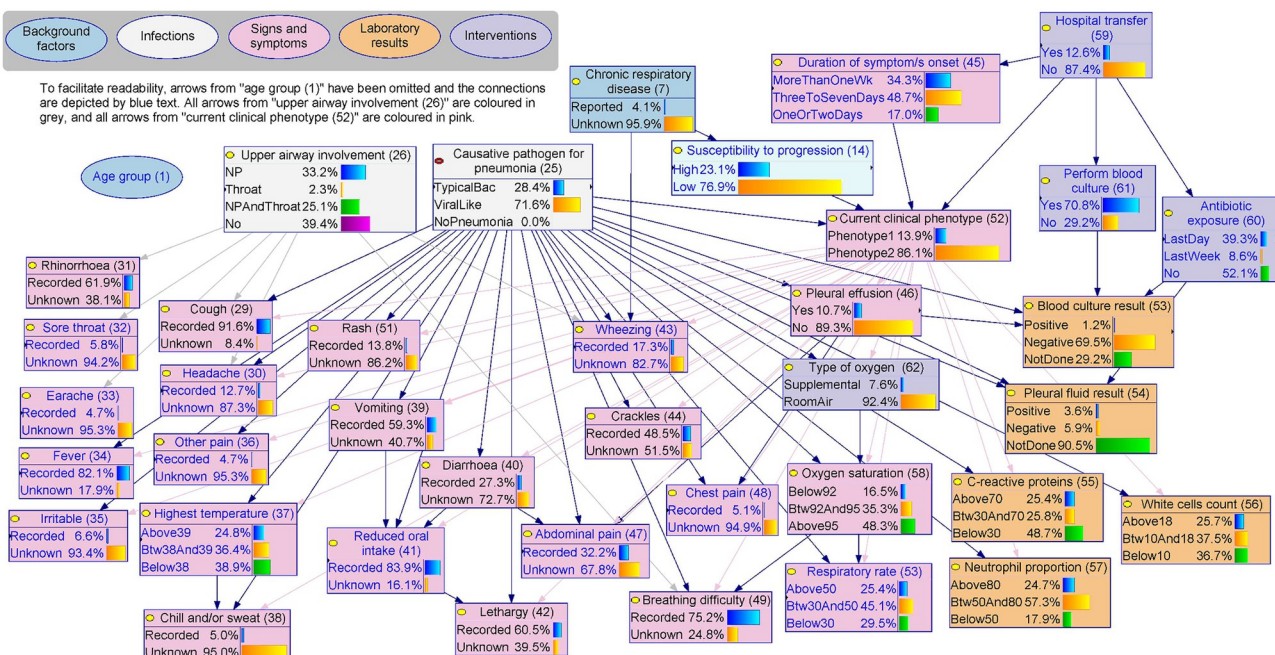

**Fig 6. The evidence, BN v4.5.** Causative pathogen for pneumonia, current clinical phenotype and upper airway involvement interact to give rise to the various types of evidence including clinical signs, symptoms, and laboratory results. The BN dictionary (S1 Table) details how these latent variables may give rise to each type of evidence. Source model files can be accessed via Open Science Framework (https://osf.io/m97vb/).

interact to give rise to the various types of evidence including clinical symptoms, signs, and laboratory results (Fig 6). The potential causes of signs, symptoms, and laboratory results are rarely deterministic, singular or mutually exclusive. For example, **cough (29)** can be driven by upper airway irritation, inflammation, or mucus production (e.g., postnasal drip secondary to rhinorrhoea/rhinitis) or can be from pneumonia affecting the lower airways. Similarly, increased **respiratory rate (53)** can be a consequence of airway inflammation or low **oxygen saturation (58)** and can vary by the child's age.

Background (dark blue) and intervention (purple) factors may also influence certain types of evidence. For example, age is an important determinant of the report of pain (e.g., **headache (30)**, **abdominal pain (47)** and **sore throat (32)**) because children need to be old enough to communicate these symptoms. Recent **antibiotic exposure (60)** may reduce the probability of a successful blood culture (**blood culture result (53)**) even if a child is bacteremic. Knowing the **type of oxygen (62)** (*supplemental* or *room air*) is essential for interpreting oxygen saturation in relation to lung function. Symptoms and signs may also interact, e.g., **vomiting (39)** and **diarrhoea (40)** may lead to **reduced oral intake (41)** which could subsequently lead to **lethargy (42)**.

The classification of the current clinical phenotype of pneumonia is analogous to a count-based metric for disease severity, where those with *phenotype 1* had a median of 12 pneumonia-relevant signs and symptoms, compared to a median of 8 for *phenotype 2*. The inclusion of this variable allowed us to separate the differential influence of causative pathogen (typical bacterial or viral-like) from the severity of clinical manifestations conditional on the causative pathogen. For example, the overall probability of a child being *phenotype 1* (severe) was almost identical for pneumonia involving viruses and viral-like pathogens (14.0%) and those involving typical bacterial pathogens (13.6%). For a child who had symptom onset within two days

and been transferred from another hospital, the probability of *phenotype 1* was much higher for viral-like pneumonia (79.4%) than typical bacterial case (42.6%). In contrast, if the child presented to ED directly (not transferred), the overall probability of *phenotype 1* is lower (15.5%), however, typical bacterial pneumonia now has a more severe phenotype (23.8%) than their viral-like counterparts (12.1%).

### 3.3 Model evaluation and sensitivity analyses

A child's causative pathogen for pneumonia (25) is our ideal target for prediction because it would respond (or not respond) to antibiotic use and subsequently drive the progression of disease. While the model can predict this variable, it is latent, making it challenging to evaluate using observational data alone. Hence, we instead assess the performance of the BN using the recorded clinically-confirmed bacterial pneumonia (28) as the best estimate of the causative pathogen type, acknowledging that some of these diagnoses may be incorrect. So long as the number of incorrect diagnoses is not great (less than 5% in BN v4.5 would certainly be acceptable), this will not affect our conclusions. More specifically, the assumption that node 28 is a good surrogate would not hold if 'viral-like pneumonia (node 25)' can also probabilistically lead to pleural effusion or positive blood culture, and in BN v4.5, these probabilities are very low: 3% and 0% respectively. Fig 7 presents the BN predicted probabilities of clinically-confirmed bacterial pneumonia for every pneumonia episode for children in the cohort, and compares these against their final diagnosis. The graphs represent four input scenarios *(a-d)*, each providing more information to the model than its preceding scenario, which approximately replicate how information may chronologically become available in clinical practice, serving also as examples of potential time points at which the BN can be applied for decision support:

*(a)* provides the model with available basic background factors of the partipants; these can include age group, childcare days per week, chronic respiratory disease, ethnicity, influenza season, prematurity, smoker in the household, impaired immunity, influenza vaccine, pneumococcal vaccine, and previous significant infection.

*(b)* uses the information in *(a)* plus signs, symptoms and interventions if reported; these can include duration of symptom/s onset, rhinorrhoea, cough, respiratory rate, oxygen saturation, breathing difficulty, crackles, wheezing, (parent-reported) fever, highest temperature, chill, sweats, irritability, vomiting, diarrhoea, reduced oral intake, lethargy, chest pain, abdominal pain, other pain, earache, sore throat, headache, rash, (prior) antibiotic exposure, type of oxygen, hospital transfer, and (decision to) perform blood culture.

*(c)* uses information in *(b)* plus any detection (or no detection) of respiratory pathogens in the nasopharynx; these can include RSV, influenza, HMPV, parainfluenza, mycoplasma, and any typical bacteria.

*(d)* includes *(c)* plus all other available results; these can include C-reactive proteins (CRP), white cell count (WCC), and neutrophil proportion (of total WCC).

Graphs that show a strong overlap (such as in Fig 7a) indicate the model is only able to achieve weak discrimination between bacterial diagnoses and others. However, as further information is provided (through scenarios (b), (c) and (d)), the discrimination improves, as evidenced by both the graphs and the improving AUROC scores. In contrast, the log score does not improve over the first 3 scenarios; coupled with the improving AUROC, this suggests the initial probabilistic prediction is already quite close to the true class. Log loss does improve in the final scenario, as does AUROC again and the greatest discrimination is visible in this

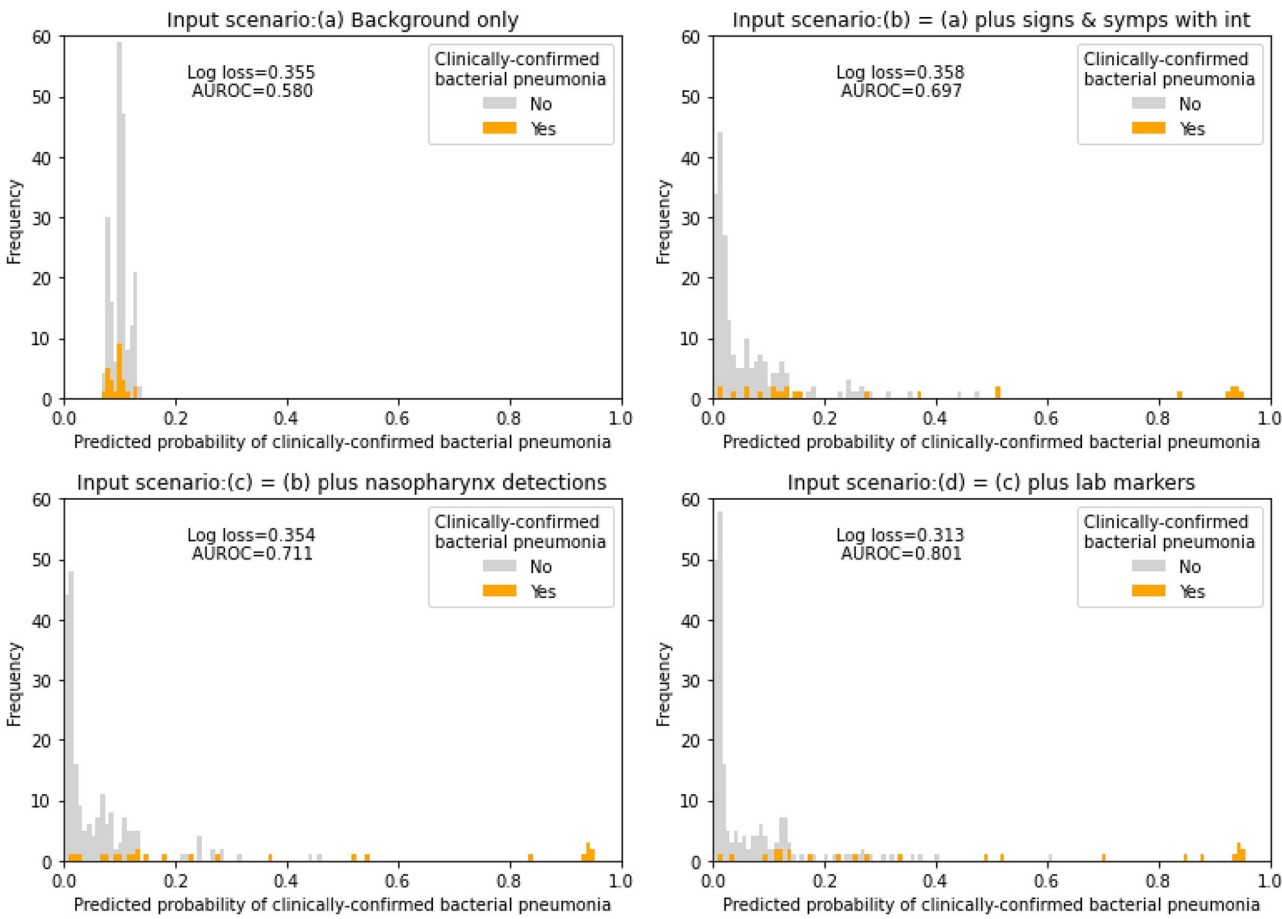

**Fig 7. The performance of BN v4.5.** Predicted probabilities of clinically-confirmed bacterial pneumonia are compared with the final observations (orange if yes) for four input scenarios, with mean log loss and AUROC based on 10-fold cross-validation. For each scenario, available information from each participant was entered as input for the BN to predict the clinically-confirmed bacterial pneumonia.

graph. Table 1 additionally presents the evaluation metrics for other observable targets that are also close surrogates of the (latent) causative pathogen for pneumonia (the primary target for prediction), including pleural effusion, positive blood culture result, and the detection of viruses or mycoplasma in the nasopharynx. As expected we see improved performance in predicting these variables as more observations are entered in the input scenarios, confirming that the model is likely to behave in a reasonable way when predicting the primary target.

From a clinical decision-making perspective, a model needs to provide predictions that trade-off false positive and false negative rates (FPR and FNR, respectively). More specifically, such a trade-off needs to be defined under a particular decision-making context or, in other words, alternative FPR/FNR trade-off preferences. Table 2 shows how the desirable threshold in predicting a target variable may vary greatly under different input scenarios and different FPR/FNR trade-off preferences—i.e., different relative weightings of the consequences of false positive versus false negative predictions. For example, when FPR and FNR are weighted equally, lower thresholds help to simultaneously minimise FPR and FNR when more information becomes available for input scenarios (c) and (d). Under the same input scenarios, the preferred threshold decreases further as higher weight is assigned to FNR.

**Table 1. The performance of BN v4.5 in predicting a range of evaluation targets under different input scenarios, in the form of mean log loss and AUROC based on 10-fold cross-validation.** Note when the prediction targets themselves were included as observed variables in the corresponding input scenario, we removed them from the list of input variables. For example, no prediction about RSV in nasopharynx under input scenario (c) is made (and hence shown as NA) because RSV status is already known in that scenario. Raw results and scripts for generating these metrics can be accessed via Open Science Framework (https://osf.io/m97vb/).

| Evaluation target | AUROC | | | | Log loss | | | |
|---|---|---|---|---|---|---|---|---|
| *Input scenario* | *(a)* | *(b)* | *(c)* | *(d)* | *(a)* | *(b)* | *(c)* | *(d)* |
| **Clinically-confirmed bacterial pneumonia (28)** | 0.58 | 0.70 | 0.71 | 0.80 | 0.36 | 0.36 | 0.35 | 0.31 |
| **Pleural effusion (46)** | 0.58 | 0.73 | 0.73 | 0.81 | 0.35 | 0.33 | 0.33 | 0.30 |
| **Positive blood culture (53)** | 0.73 | 0.74 | 0.80 | 0.91 | 0.67 | 0.08 | 0.08 | 0.06 |
| **RSV in nasopharynx (16)** | 0.74 | 0.75 | NA | 0.75 | 0.48 | 0.51 | NA | 0.52 |
| **Influenza in nasopharynx (15)** | 0.52 | 0.56 | NA | 0.56 | 0.29 | 0.28 | NA | 0.28 |
| **HMPV in nasopharynx (17)** | 0.53 | 0.54 | NA | 0.54 | 0.36 | 0.36 | NA | 0.36 |
| **Parainfluenza in nasopharynx (18)** | 0.45 | 0.49 | NA | 0.52 | 0.25 | 0.25 | NA | 0.24 |
| **Mycoplasma in nasopharynx (19)** | 0.67 | 0.73 | NA | 0.73 | 0.33 | 0.32 | NA | 0.32 |

We varied relevant parameters by 20% to investigate how various factors affect the predicted probability of **typical bacterial pneumonia (24)** which is deterministically related to the *bacterial* state of the **causative pathogen for pneumonia (25)** (Fig 8). Three groups of variables were investigated, and the target prediction is most sensitive to parameters describing the pathogenicity of the potential pathogens in causing pneumonia (white), ranging from 12% to 50%. Varying the prevalence of various pathogens in the nasopharynx caused the predicted probability of the involvement of typical bacteria to vary from 22% to 30% (yellow). Varying

**Table 2. The optimal performance of the BN in predicting clinically-confirmed bacterial pneumonia (28) under different input scenarios and different FPR/FNR trade-off preferences (P1-P4).** Specifically, we defined *optimal* based on a minimised weighted sum of FPR and FNR across the 10-fold cross-validation results. For each input scenario and FPR/FNR trade-off preference, we present here the threshold used when the optimal performance is achieved, and its associated false positives (FP), false negatives (FN), sensitivity and specificity. Note that there is a total of 25 positive observations of clinically-confirmed bacterial pneumonia in our cohort of 230 cases of X-ray confirmed pneumonia. Raw results and scripts for generating these metrics can be accessed via Open Science Framework (https://osf.io/m97vb/).

| Evaluation metric \ *Input scenario* | | *(a)* | *(b)* | *(c)* | *(d)* |
|---|---|---|---|---|---|
| P1: FPR and FNR are equally weighted | threshold (>) | 0.069 | 0.096 | 0.058 | 0.035 |
| | FP | 174 | 39 | 57 | 69 |
| | FN | 2 | 11 | 9 | 3 |
| | sensitivity | 92% | 56% | 64% | 88% |
| | specificity | 15% | 81% | 72% | 66% |
| P2: FNR has a weight that is 3 × FPR | threshold (>) | 0.039 | 0.006 | 0.008 | 0.035 |
| | FP | 202 | 188 | 151 | 69 |
| | FN | 0 | 0 | 1 | 3 |
| | sensitivity | 100% | 100% | 96% | 88% |
| | specificity | 1% | 8% | 26% | 66% |
| P3: FNR has a weight that is 5 × FPR | threshold (>) | 0.039 | 0.006 | 0.006 | 0.013 |
| | FP | 202 | 188 | 187 | 121 |
| | FN | 0 | 0 | 0 | 1 |
| | sensitivity | 100% | 100% | 100% | 96% |
| | specificity | 1% | 8% | 9% | 41% |
| P4: FNR has a weight that is 10 × FPR | threshold (>) | 0.039 | 0.006 | 0.006 | 0.005 |
| | FP | 202 | 188 | 187 | 185 |
| | FN | 0 | 0 | 0 | 0 |
| | sensitivity | 100% | 100% | 100% | 100% |
| | specificity | 1% | 8% | 9% | 10% |

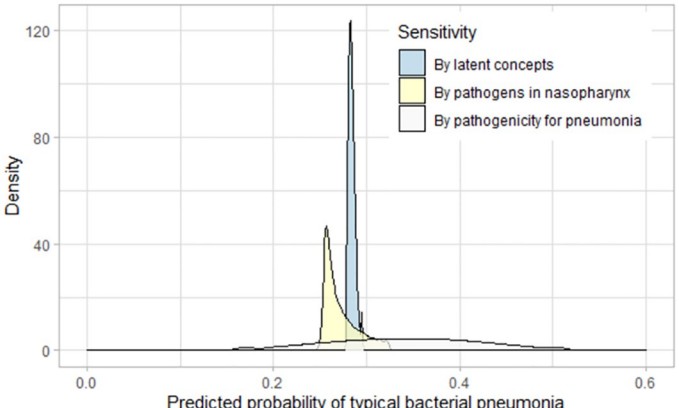

**Fig 8. Sensitivity analysis, BN v4.5.** Relevant parameters were varied by 20% to investigate how they affected the predicted probability of involvement of typical bacteria. This target prediction was most sensitive to the pathogenicity of potential pathogens in causing pneumonia (white), followed by the prevalence of various pathogens in nasopharynx (yellow), and was least sensitive to the latent epidemiological concepts (blue). CPTs that were varied in the pathogenicity group include those for both **viral-like pneumonia (23)** and **typical bacterial pneumonia (24)**. The pathogens in nasopharynx group involve CPTs for **RSV in nasopharynx (16)**, **Influenza in nasopharynx (15)**, **HMPV in nasopharynx (17)**, **Parainfluenza in nasopharynx (18)**, **Mycoplasma in nasopharynx (19)** and **bacteria in nasopharynx (20)**. And the latent concepts group involved CPTs for **level of exposure (12)**, **susceptibility to colonisation (13)** and **susceptibility to progression (14)**.

the assumptions regarding the latent epidemiological concepts (e.g., level of exposure) had a small influence on the probability of involvement of typical bacteria (25% to 27%) (blue).

The use of KL divergence analysis (S2 Table) allowed us to highlight the discrepancies between prior and posterior CPTs in particular among the hybrid expert-data variables; *i.e.*, which expert-informed CPTs were shifted the most by the data. We noticed that both **viral-like pneumonia (23)** and **typical bacterial pneumonia (24)** rank highly among the hybrid expert-data variables. As an interesting example, given the presence of a nasopharynx infection, but negative detections for all modelled viral-like pathogens in the nasopharynx, the probability of a viral pneumonia still shifts from 11% to 98% (keeping in mind the model is being asked to predict a pneumonia of some sort). This implies a potentially more significant role for other respiratory viruses among children with X-ray confirmed pneumonia.

## 3.4 Demonstrative clinical scenarios

When serving as a clinical decision support tool, the available background factors, clinical observations and investigation results would be entered as model inputs for making inferences about **causative pathogen for pneumonia (25)** in a particular episode. We presented three scenarios to demonstrate the potential use of this model in a clinical context Figs (9)–(11). Proposed and reviewed by domain experts, the selected variables in these scenarios are not those you would specifically measure or look for when suspecting bacterial pneumonia—rather this is information commonly present when clinicians need to make a diagnosis of bacterial pneumonia and decide on interventions such as antibiotic use.

The BN predicted probability of typical bacterial pneumonia is 28% for the cohort of children with X-ray confirmed pneumonia (labelled in the blue box at the top right corner). For each scenario, this prediction is updated, branching in multiple possible ways that are conditional on information that may become available over time along the child's clinical course and thus act as potential time points when the BN can be applied to make predictions for

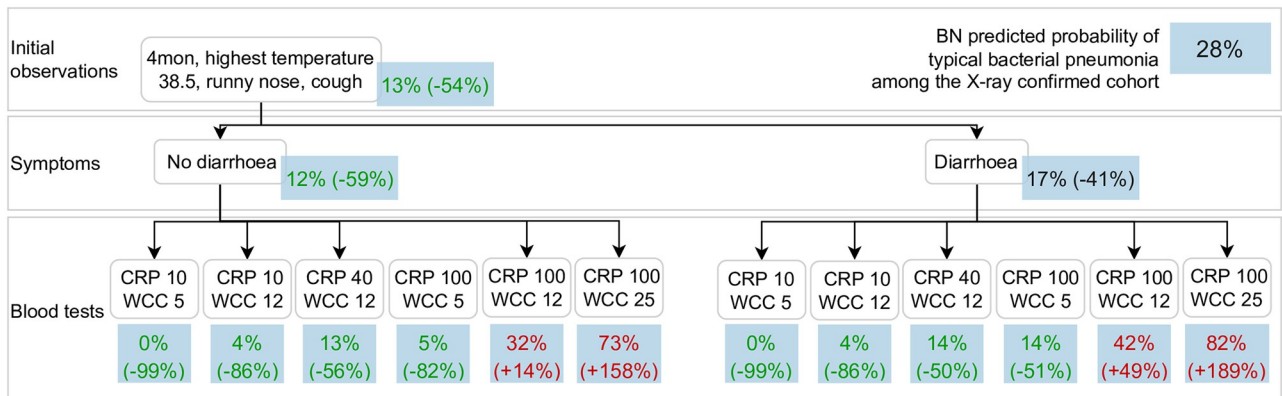

**Fig 9. Clinical scenario 1, BN v4.5.** This use case presents an infant (**age group (1)**) with X-ray confirmed pneumonia, moderately high temperature (**highest temperature (37)**), **cough (29)** and runny nose (**rhinorrhoea (31)**). The BN predicts a low probability of typical bacterial pneumonia for this child (**causative pathogen for pneumonia (25)**), and a reported history of **diarrhoea (40)** alone does not largely affect the prediction. The target prediction remains low unless both the **C-reactive proteins (55)** and **white cells count (56)** are moderately elevated.

decision support. In parentheses we provide the change in the target prediction in relation to the cohort average; we use red text to indicate if the predicted probability of typical bacterial pneumonia is higher than the cohort average, and use green text to indicate predicted target probabilities which are at least 50% lower than the cohort average.

Fig 9 presents a scenario in which the BN predicts a low probability (13%) of a typical bacterial cause of pneumonia for a 4 month old infant with X-ray confirmed pneumonia with moderate signs and symptoms (i.e., highest temperature of 38.5 degrees Celcius, runny nose and cough). The predicted probability remains low unless both the CRP and WCC are high (at least 100 $mg/L$ and $12 \times 10^9/L$, respectively). In contrast, if instead a 9 year old (school-aged) child with X-ray confirmed pneumonia presents various signs and symptoms (Fig 10), the probability of typical bacterial pneumonia is predicted to be higher and similar to the cohort average. The predicted probability of typical bacterial pneumonia is almost doubled if the

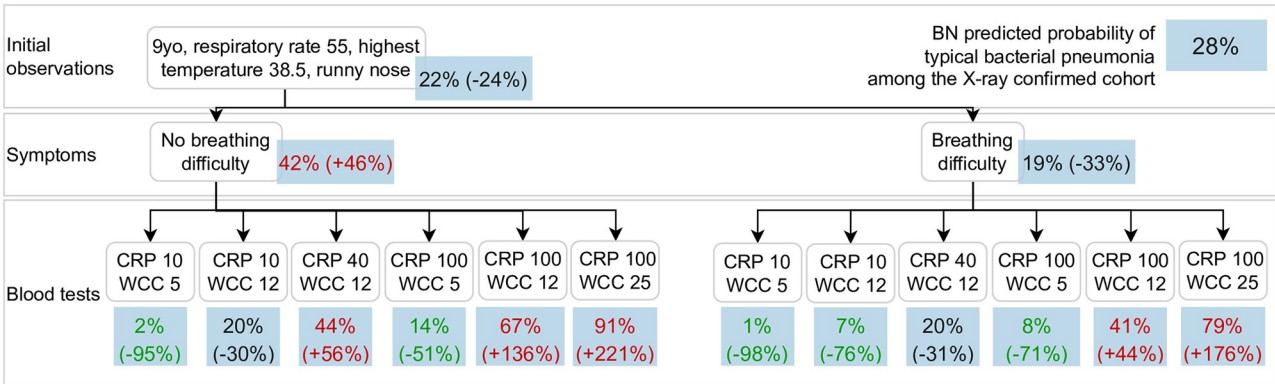

**Fig 10. Clinical scenario 2, BN v4.5.** This scenario presents a school-aged (**age group (1)**) child with X-ray confirmed pneumonia and with moderately high temperature (**highest temperature (37)**), runny nose (**rhinorrhoea (31)**) and high **respiratory rate (53)**. The BN predicts a 22% probability of typical bacterial pneumonia for this child (**causative pathogen for pneumonia (25)**), and this prediction increases to 42% if the child has no **breathing difficulty (49)**. Moderate and very high **C-reactive proteins (55)** and **white cells count (56)** further increase the probability of a typical bacterial cause for this pneumonia.

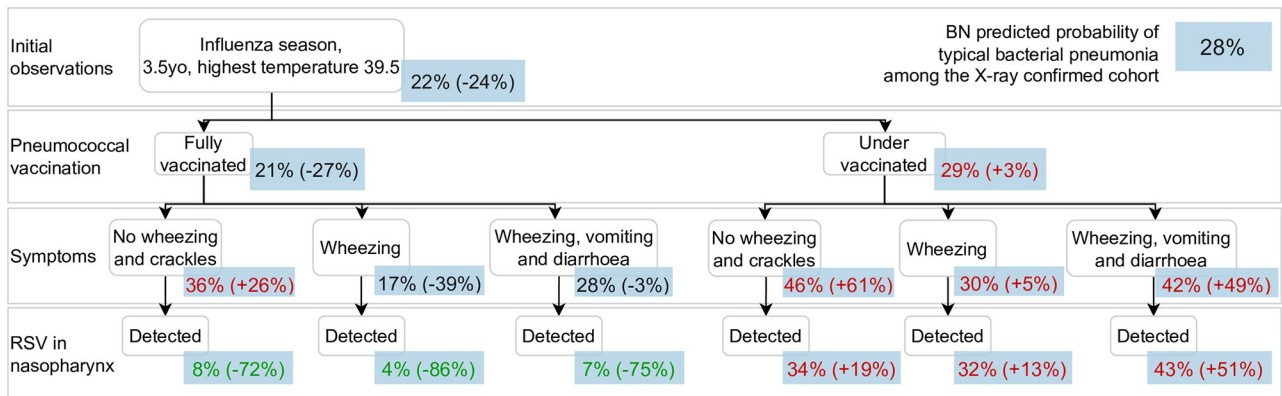

**Fig 11. Clinical scenario 3, BN v4.5.** This scenario shows a pre-school aged (**age group (1)**) child with X-ray confirmed pneumonia and very high temperature (**highest temperature (37)**) during the **influenza season (9)**. The probability of typical bacterial pneumonia (**causative pathogen for pneumonia (25)**) increases if the child is under-vaccinated for pneumococcus (**Pneumococcal vaccine (10)**), regardless of whether RSV is detected from their nasopharynx or not (**RSV in nasopharynx (16)**). If fully-vaccinated, the presence of **wheezing (43)** will lead to a decrease in the predicted probability of typical bacterial pneumonia, and this prediction will drop to only 4% if RSV is detected in the child's nasopharynx. Other variables involved in this scenario are **crackles (44)**, **vomiting (39)** and **diarrhoea (40)**.

child has no breathing difficulty, and a moderately high CRP (40 $mg/L$) can be indicative for a bacterial cause. In other words, the significance of a raised CRP varies under various scenarios. The third scenario (Fig 11) presents a 3.5 year old (pre-school aged) child with X-ray confirmed pneumonia and very high temperature during the influenza season. The probability of typical bacterial pneumonia increases if the child is under-vaccinated for pneumococcus, regardless of whether RSV is detected in their nasopharynx. If fully-vaccinated, the presence of wheezing will lead to a decrease in the predicted probability of typical bacterial pneumonia, and this prediction will drop to only 4% if RSV is detected in the child's nasopharynx.

## 4 Discussion

The diagnostic challenge of differentiating bacterial from non-bacterial pneumonia is the main driver of antibiotic use for treating pneumonia in children. We have built a causal BN that captures the complex epidemiological context of paediatric pneumonia and how this gives rise to the variable clinical manifestations observed in the hospital setting. Based on the combined use of domain expert knowledge and data, the model was designed to apply to a cohort of children with X-ray confirmed pneumonia who presented to a tertiary paediatric hospital in Australia. The resulting BN offers explainable and quantitative predictions on a range of variables of interest, including the causative pathogen for pneumonia, the clinically-confirmed bacterial pneumonia, the detection of respiratory pathogens in the nasopharynx, and the clinical phenotype of a pneumonia episode. Because the BN is causal, it serves as a starting point to integrate the effects of further preventative or interventional measures, potentially leading to future insights into how public health strategies may drive the clinical epidemiology of pneumonia, or how clinical antibiotic use may alter the disease progression of a given episode.

We used AUROC and log loss to quantitatively assess the model's performance in predicting the directly observable variables like the diagnosis of pleural effusion, positive blood culture and detection of RSV and mycoplasma in the nasopharynx. Based on 10-fold cross-validation, the model performance steadily improved as more information was entered (Table 1). We specifically highlight that a desirable quantitative threshold for practical use is very dependent upon different input scenarios and FPR/FNR trade-off preferences, providing

insight into how computational model predictions may be translated to actionable decisions in practice. Qualitatively, we used commonly encountered scenarios to assess the potential usefulness of the model outputs in various clinical pictures. To our knowledge, this is the first causal model developed to help determine the causative pathogen for paediatric pneumonia. Our model framework and the methodological approach can be adapted beyond our context to broad respiratory infections and geographical and healthcare settings.

## 4.1 Using latent concepts to reveal the epidemiological context

When a child has X-ray confirmed pneumonia, the causative pathogen for pneumonia usually remains unobserved, i.e., latent. Patient background factors such as basic demographics, seasonality, vaccination and comorbidities can help predict the latent causative pathogen because they often causally influence the presence of a respiratory pathogen in the child's nasopharynx or respiratory tract which in turn predisposes to pneumonia. For example, RSV is more prevalent in the winter in temperate environments, owing to its elevated circulation in the community, and bacterial infection is more probable in a younger child due to the interplay between their developing immunity and respiratory microbiota [44, 45]. The difficulties faced by clinicians arises from the dynamic and complex interactions amongst these factors, as we saw in Fig 11 for a young child who presents to hospital in the influenza season.

We addressed this challenge by firstly organising relevant variables under a causal framework (Fig 4), and introducing several latent concepts such as the level of exposure and the susceptibility of a child to colonisation. This approach helped explain the potentially complex role of each variable in the problem domain. For example, older children may have lower susceptibility to bacterial colonisation as well as to pneumonia progression, with both leading to a reduced probability of typical bacterial pneumonia [45]. However, they are also more likely to have a high exposure through attendance at childcare or school [46], leading to an increase in the nasopharyngeal presence of all pathogens, thereby increasing the potential for subsequent infection. Being explicit about these underlying mechanisms allowed us to capture and understand the role of each factor in the context of each pneumonia episode, which in turn gave us further insight into the epidemiological context and more precise predictions of the causative pathogen for pneumonia.

## 4.2 Mapping clinical and laboratory evidence to explicit causal paths

Model predictions can be further improved by utilising additional information which is likely to be available shortly after presentation, namely, clinical observations and laboratory investigation results. By modelling the overlap and divergence in the pathophysiological pathways indicated by each piece of clinical and laboratory evidence (Fig 6), the causal BN approach can estimate the episode-specific probability of each causative pathogen based on any input variables available. As illustrated in clinical scenario 1 (Fig 9), while diarrhoea can be a feature of either viral or bacterial pneumonia, knowing about diarrhoea alone does not greatly affect the predicted probability of typical bacterial pneumonia. However, when additional information such as CRP and WCC becomes available, the prediction can vary from close to 0% up to 82%.

A challenge faced by clinicians is the variable predictive value of specific clinical and laboratory observations when the underlying epidemiology changes [15]. More specifically, typical bacterial pneumonia is more likely to drive elevated WCC and CRP [35], however, elevated biomarkers can also be caused by viral pathogens, especially when disease manifestation is severe, leading to reduced specificity for bacterial infection. Our BN addressed this issue by introducing a variable to summarise the clinical phenotype for each episode, which functioned to separate the influence of the causative pathogen from the clinical progression of the

presenting episode based on the patient's susceptibility to progression and the duration and severity of the presenting illness. This was illustrated in Scenario 1 (Fig 9) and Scenario 2 (Fig 10), in which the same blood results could lead to significantly different predictions of typical bacterial pneumonia for the two demographically and clinically distinct presenting episodes.

### 4.3 The challenge and need to better understand pathogenicity

It became apparent since early in the model development (after Workshop 1) that the pathogenicity of different organisms plays a central role in driving the probability that a pathogen is causative. There are a large number of organisms (viral and bacterial) that are capable of causing pneumonia, while there are also frequent detections of multiple potentially pathogenic organisms in the nasopharynx. Hence, it is important to understand the prevalence of relevant organisms in the respiratory tract (including the nasopharynx) as well as the probability of being causative of pneumonia given they are present there. These issues affect how the model should be structured (e.g., to what extent we should separate or group pathogens) and how we should parameterise the model (e.g., given the co-presence of more than one pathogen, which one is the more likely to be the causative agent). Many of these issues remain the subject of debate and ultimately unresolved by the scientific community [47].

We designed a series of survey questions to elicit the current understanding and knowledge of domain experts (Survey 2, 3 and 5, see S1 File, Section C, D and F). Finding the best way to frame questions proved challenging given the complexity of the problem domain. Consequently, we explored a variety of approaches based on modelling needs, facilitated by graphical illustration, data visualisation, and various question formats. This process also helped clarify the modelling requirements themselves, helping us to better manage our current imperfect understanding of the problem domain. In the areas of greatest need, sensitivity analyses were subsequently conducted to understand how much the uncertainty in our understanding could impact on the BN predicted outcomes (Fig 8).

### 4.4 Implications for providing decision support for antibiotic prescription

Among the 230 children enrolled in our cohort of X-ray confirmed pneumonia, 47% were exposed to recent antibiotic use prior to the point of hospital admission when baseline study data were collected. Of these, 99% eventually received antibiotics during their hospital stay (post-enrolment), although only 11% had clinically-confirmed bacterial pneumonia [35]. This suggests that clinicians rarely have the information they need to have confidence to withhold antibiotics. Applied to this cohort, the resulting BN predicted that 28% had typical bacterial pneumonia (either bacterial or viral-bacterial co-infection), indicating the potential to greatly rationalise use of antibiotics. Nevertheless, translating mathematical predictions into better clinical decision-making is not straightforward, as the the best practical use of a model is affected by its ability to predict the real world, i.e., by its performance characteristics. To this end, we showed the most desirable model output threshold for choosing an action can be very sensitive to different input scenarios and trade-off preferences (Table 2).

From a clinical decision making perspective, the choice of model threshold for guiding antibiotic use should take into account the *values* (utilities or preferences) that one attaches to possible consequences of their decision, e.g., how much clinicians are willing to accept false positives (i.e., over-treating a non-bacterial infection with antibiotics) in order to avoid false negatives (i.e., missing/delaying antibiotic therapy for a bacterial infection). Although satisfactory numeric predictive performance has been achieved by our BN in predicting directly observable variables, including an AUROC of 0.8 in predicting clinically-confirmed bacterial pneumonia with sensitivity 88% and specificity 66%, we can only assess the acceptability of

these performance characteristics within the context of specific input scenarios and trade-off preferences (i.e., preferences over the trade-off between FPR and FNR). Looking at Table 2 on this basis, with the clinical data corresponding to FPR/FNR trade-off preference P3, in column (a) the model predicts a positive result in 202 of 205 cases that were not clinically-confirmed bacterial pneumonia. Hence, almost all the truly negative cases would receive antibiotics, suggesting little additional value over current clinical practice (recalling that in our cohort, 99% received antibiotics). Conversely, column (d) shows that, with more information in the input scenario, the model predicts a negative result for 84 of the 205 cases without clinically-confirmed bacterial pneumonia, suggesting the model may help avoid up to 41% of unnecessary antibiotic prescriptions. Of note, clinicians may also want to adjust their FPR/FNR trade-off preference based on a specific patient's circumstances. For example, they may reduce the acceptable FNR for subgroups that are more vulnerable to potentially severe consequences of a false negative result for bacterial pneumonia (such as neonates); or they may accept a higher FNR for older children where close follow-up is possible.

The model's behaviour in predicting the (latent) causative pathogens for pneumonia was qualitatively reviewed in various forms, including in Workshops 3 and 4 and Survey 4 (S1 File, Section E). The review outcomes from our experts from diverse domain areas provided confidence in the potential clinical value that this BN offers as a decision support tool. The three scenarios in §3.4 provide an insight into how we envisage the model outputs could support diagnostic decisions under various clinical contexts, leading to numerous questions about how model outputs should be presented. Even if causal models have good performance and produce explainable outputs, they will not be useful tools unless their outputs can be presented in a form that end-users can understand, interpret and use. The barriers to implementation and uptake that decision support tools including BNs face are widely reported [48, 49].

## 4.5 Study limitations and future direction

The BN needs further validation before it can be clinically implemented, particularly examining the performance of the model in additional cohorts that differ in both key contextual factors and in the availability and operational definitions of superficially equivalent variables. Examples of such differences include: whether the definition of pneumonia is distinct across studies; whether a different spectrum of respiratory infections are involved; and how the clinical epidemiology differs based on the geographically distinct transmission dynamics or variations across healthcare systems. In particular, we highlight the important distinction between variable definitions (or in this case a BN dictionary) and a data dictionary. A variable definition would (usually) not change if it has been well designed to describe a certain mechanistic process in a problem domain; such a variable could then be readily *mapped* to an external dataset that intends to capture observations about the very same mechanistic process. Any significant discrepancy between a variable definition and an associated data definition should lead to a review and potential revision of the model structure with respect to the original modelling purpose, with the aim to improve the adaptability of the model, the interpretation of model performance once adapted, and the development of the BN for future adaptation and implementation. Several measures were taken to address the overfitting problem, including the 10-fold cross-validation, expert-informed priors and the KL divergence assessment. We acknowledge that the strongest test of overfitting must make use of an external dataset; however, the KL divergence results provide a unique insight into where such overfitting may be present in the model, providing a clear focus for any future evaluation that assesses the model on external datasets.

It is important to note there is a distinction between the actual presence of an organism at a body site and its successful detection by sampling at that body site; we only ever know about the former via evidence of the latter. In the model, we used the detection of organisms as a surrogate for their actual presence in the nasopharynx, in order to reduce the number of latent variables. For our model development, these data were collected prospectively and PCR testing was performed systematically to identify relevant pathogens over and above the routine clinical practice, generating high quality information with little missing data. However, the quality of model predictions will not be as robust if data from routine clinical review and investigation are used alone. Future work should consider the sensitivity and specificity of the laboratory techniques used to capture this information.

The BN can be further improved using richer data with a higher level of granularity. For example, information on the severity of reported symptoms was limited. We would like to further improve the classification of pneumonia (or reduce potential mis-classification due to potential limitations of assessment techniques such as X-rays) by capturing a more comprehensive spectrum of lower respiratory tract infections such as bronchitis and bronchiolitis, and in particular how they should be explicitly mapped to other model variables. Furthermore, other comorbidities were identified but are yet to be included in the model, such as the history of neurological conditions that may affect susceptibility to colonisation and disease progression. There is also the potential to include further information on the use of intravenous fluid, intensive care management and patient progression over time, which would enable the model to predict the likely effects of specific interventions. Integrating such models into electronic medical record systems to enable efficient data capture is an important future direction, as it would significantly improve the ability to evaluate and eventually trial, maintain and use these models and, ultimately, to derive the most value from them.

## Supporting information

**S1 File. Project schematic and survey questions.** This file includes a detailed schematic of the project, as well as a summary of survey questions.
(PDF)

**S1 Table. BN v4.5 dictionary.** In this BN dictionary, we provide information on each model variable, including identifiers, what it means and how it's mapped to data, how it's discretised in the BN, its parents in the BN, and how it's mechanistically affected by those parent nodes. The dictionary also notes the observational status (whether observable, latent or derived), and the parameterisation method used (data-driven, expert-data, or expert-driven), as well as a high-level category.
(XLSX)

**S2 Table. KL comparison.** In this document, we provide the KL divergence assessment results that compare the difference between the prior and posterior CPTs of BN v4.5 (by node on tab 1 and by row on tab 2), before and after training with data. The full CPTs as well as script for conducting this analysis can be accessed via Open Science Framework (https://osf.io/m97vb/).
(XLSX)

## Author Contributions

**Conceptualization:** Yue Wu, Steven Mascaro, Thomas L. Snelling, Christopher C. Blyth.

**Data curation:** Yue Wu, Mejbah Bhuiyan, Michael Dymock.

**Formal analysis:** Yue Wu, Steven Mascaro, Parveen Fathima, Michael Dymock.

**Funding acquisition:** Yue Wu, Steven Mascaro, Mejbah Bhuiyan, Mark P. Nicol, Julie A. Marsh, Thomas L. Snelling, Christopher C. Blyth.

**Investigation:** Yue Wu, Steven Mascaro, Mejbah Bhuiyan, Parveen Fathima, Ariel O. Mace, Mark P. Nicol, Peter C. Richmond, Lea-Ann Kirkham, David A. Foley, Charlie McLeod, Meredith L. Borland, Andrew Martin, Phoebe C. M. Williams, Thomas L. Snelling, Christopher C. Blyth.

**Methodology:** Yue Wu, Steven Mascaro, Mejbah Bhuiyan, Parveen Fathima, Ariel O. Mace, Mark P. Nicol, Peter C. Richmond, Lea-Ann Kirkham, David A. Foley, Charlie McLeod, Meredith L. Borland, Andrew Martin, Phoebe C. M. Williams, Julie A. Marsh, Thomas L. Snelling, Christopher C. Blyth.

**Project administration:** Yue Wu, Mejbah Bhuiyan.

**Resources:** Yue Wu, Mejbah Bhuiyan.

**Software:** Yue Wu, Steven Mascaro.

**Supervision:** Thomas L. Snelling, Christopher C. Blyth.

**Validation:** Yue Wu, Steven Mascaro.

**Visualization:** Yue Wu, Steven Mascaro, Michael Dymock.

**Writing – original draft:** Yue Wu, Steven Mascaro.

**Writing – review & editing:** Mejbah Bhuiyan, Parveen Fathima, Ariel O. Mace, Mark P. Nicol, Peter C. Richmond, Lea-Ann Kirkham, Michael Dymock, David A. Foley, Charlie McLeod, Meredith L. Borland, Andrew Martin, Phoebe C. M. Williams, Julie A. Marsh, Thomas L. Snelling, Christopher C. Blyth.

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
