## [Decision Letter · Decision Letter 0]

16 Nov 2022

Dear Dr Wu,

Thank you very much for submitting your manuscript "Predicting the causative pathogen among children with pneumonia using a causal Bayesian network" for consideration at PLOS Computational Biology.

As with all papers reviewed by the journal, your manuscript was reviewed by members of the editorial board and by several independent reviewers. In light of the reviews (below this email), we would like to invite the resubmission of a significantly-revised version that takes into account the reviewers' comments.

I agree with both reviewers that this is an interesting study that is likely to be of interest to the computational biology community. I also agree with both reviewers that more clarification around the methodology and interpretation is needed and that claims of clinical relevance should be tempered. I also believe the authors need to provide additional details around the data to be in compliance with our data availability policy.

We cannot make any decision about publication until we have seen the revised manuscript and your response to the reviewers' comments. Your revised manuscript is also likely to be sent to reviewers for further evaluation.

Sincerely,

Samuel V. Scarpino

Academic Editor

PLOS Computational Biology

Virginia Pitzer

Section Editor

PLOS Computational Biology

I agree with both reviewers that this is an interesting study that is likely to be of interest to the computational biology community. I also agree with both reviewers that more clarification around the methodology and interpretation is needed and that claims of clinical relevance should be tempered. I also believe the authors need to provide additional details around the data to be in compliance with our data availability policy.

Reviewer's Responses to Questions

**Comments to the Authors:**

Reviewer #1: This manuscript describes a methodology to find causal pathogens for pediatric pneumonia. The study was carried out by surveying multiple doctors (expert knowledge) and use their knowledge along statistical methods to describe a causation DAG for the aforementioned disease which was later tested (AUROC and log-loss) using data obtained from 230 children admitted in a hospital.

The process of constructing the GAN was done iteratively using 3 distinct methods to update the nodes depending on data quality and parametrized via estimation maximization. The nodes consist in 63 total variables of multiple natures (background factors, pathogens, infection/diagnosis, etc). 3 of those variables are latent ones introduced to describe exposure, colonization and progression.

The authors use the bacterial diagnosis as a gold standard as one of the main objectives is the prediction of bacterial neumonia.

The paper is very well written. The introduction properly states the need of such study (antibiotic usage) and guides through the pneumonia topic extensively and how Bayesian Networks can have an important role in the understanding of causal effects on distinguish bacterial from non-bacterial infections.

The statistical methods were sound, relevant to the study and well described. It's refreshing to have a paper that includes domain experts in the pipeline.

I will recommend for publishing as is but I do have a some questions/comments:

Section 3.3:

“So long as the number of incorrect diagnoses is not great, this will not affect our conclusions” .

What would the authors think the threshold would be?

Nice to have an acknowledgment of the limitations, I appreciate that.

It appears to me that for the authors the importance of causality is knowing what’s the minimum data needed to correctly predict a bacterial pneumonia. It would be nice to have a discussion about the understanding of the causal effects as a preventive measure or why is it better to have a BN over a fully automated ML algorithm if the goal is the prediction of a labeled variable.

As stated in section 4.5 the flexible definition of pneumonia could be problematic although it seems that the important part is the bacterial more than the name itself. Would the scope change if the authors proceed with a larger study proposed?

Aditional file 1 was important to understand the development of the BN. Might be worthwhile to have in the main paper.

Reviewer #2: PDF attached

**Have the authors made all data and (if applicable) computational code underlying the findings in their manuscript fully available?**

Reviewer #1: **No: **The authors provided an OSF link but only the dictionary and a xdls file are provided, in the statement they said they would be sharing the source code. They said in the manuscript that they used R and Python so I was expecting a couple of files with the given extension.

Reviewer #2: **No: **The Bayesian network model has been shared together with a data dictionary (similar or same as additional file 2). The data used to train the model, in which each record would correspond to an individual, has not been shared. This is consistent with the statement made by the authors in the 'data and code availability' section. I sympathise with the authors position as, in my experience, it would likely not be possible to get permission to share this data even though it was anonymised.

PLOS authors have the option to publish the peer review history of their article (what does this mean?). If published, this will include your full peer review and any attached files.

Reviewer #1: No

Reviewer #2: No
---

## [Decision Letter · Decision Letter 1]

3 Feb 2023

Dear Dr Wu,

Thank you very much for submitting your manuscript "Predicting the causative pathogen among children with pneumonia using a causal Bayesian network" for consideration at PLOS Computational Biology. As with all papers reviewed by the journal, your manuscript was reviewed by members of the editorial board and by several independent reviewers. The reviewers appreciated the attention to an important topic. Based on the reviews, we are likely to accept this manuscript for publication, providing that you modify the manuscript according to the review recommendations.

Sincerely,

Samuel V. Scarpino

Academic Editor

PLOS Computational Biology

Virginia Pitzer

Section Editor

PLOS Computational Biology

Reviewer's Responses to Questions

**Comments to the Authors:**

Reviewer #2: 1. I am grateful to the authors for their detailed response to my comments and the updates to the paper.

2. Table 2 is a great addition to the paper. I note that the sensitivity and specificity are given as rates whereas the false positives and false negatives are numbers of cases. A consistent approach (percentages perhaps) would be preferable. Otherwise, please at least mention the dataset size (230 I think) so that the numbers can be interpreted.

3. I appreciate the new discussion in section 4.4 about the trade-off between FPR and FNR, but is it possible to be a little less abstract? My (possibly simplistic) interpretation of the clinical goal to identify true negatives (reduce unnecessary antibiotic use) without false negatives (potentially endangering patients). Looking at Table 2 on this basis, with the clinical data corresponding to column (a), the predictor is roughly useless, as 202/205 are false positives, matching current clinical practice. On the other hand, row P3 and column (d) shows the predictor avoiding approx. 40% of the unnecessary uses of antibiotics, which is potentially of benefit. I guess there may be circumstances (patients carefully observed?) in which a higher FNR is acceptable but these could be mentioned.

4. The first paragraph of the discussion section still contains the statement that ‘the model performed well’. I do not think this very general statement is justified. For example, fig 7 (d) shows that two positive cases were given very low predicted probabilities even in the input scenario with most data; in table 1 the AUROC ranges from <0.5 to 0.8. Given all the detailed attention to characterising the performance of the model (particularly in section 4.4), please could this unqualified statement be amended or removed?

**Have the authors made all data and (if applicable) computational code underlying the findings in their manuscript fully available?**

Reviewer #2: **No: **The authors have made the BN models and the evaluation results available. In similar circumstance, I do not believe I would be permitted to make the patient data available (even anonymised); I therefore accept that they are unable to do this. It might be possible for them to provide more summary statistics on the patient data, though I do not think this would add greatly.

PLOS authors have the option to publish the peer review history of their article (what does this mean?). If published, this will include your full peer review and any attached files.

Reviewer #2: No

Figure Files:

Data Requirements:

Reproducibility:

References:

---

## [Editor Report · Decision Letter 2]

22 Feb 2023

Dear Dr Wu,

We are pleased to inform you that your manuscript 'Predicting the causative pathogen among children with pneumonia using a causal Bayesian network' has been provisionally accepted for publication in PLOS Computational Biology.

Best regards,

Virginia E. Pitzer, Sc.D.

Section Editor

PLOS Computational Biology

Virginia Pitzer

Section Editor

PLOS Computational Biology

---

## [Editor Report · Acceptance letter]

8 Mar 2023

PCOMPBIOL-D-22-01147R2 

Predicting the causative pathogen among children with pneumonia using a causal Bayesian network

Dear Dr Wu,

I am pleased to inform you that your manuscript has been formally accepted for publication in PLOS Computational Biology. Your manuscript is now with our production department and you will be notified of the publication date in due course.

With kind regards,

Anita Estes
